## Invited reply

solid-state physics/materials science/physical chemistry

incommensurate composite materials, solid-state phase transitions, superspace groups, urea inclusion compounds

**Authors for correspondence:**

Michel Couzi
e-mail: michel.couzi@u-bordeaux.fr

François Guillaume
e-mail: francois.guillaume@u-bordeaux.fr

Kenneth D. M. Harris
e-mail: harriskdm@cardiff.ac.uk

†Present address: Department of Chemistry and Ilse Katz Institute for Nanoscale Science and Technology, Ben-Gurion University of the Negev, Beer-Sheva 8410501, Israel.

The accompanying comment can be viewed at http://dx.doi.org/10.1098/rsos.182073.
This article has been edited by the Royal Society of Chemistry, including the commissioning, peer review process and editorial aspects up to the point of acceptance.

# Reply to comment on Couzi *et al.* (2018): a phenomenological model for structural phase transitions in incommensurate alkane/urea inclusion compounds

Kirsten Christensen[1], P. Andrew Williams[2],
Rhian Patterson[2,3], Benjamin A. Palmer[4,†],
Michel Couzi[5], François Guillaume[5]
and Kenneth D. M. Harris[2]

[1]Inorganic Chemistry Laboratory, University of Oxford, South Parks Road, Oxford OX1 3QR, UK
[2]School of Chemistry, Cardiff University, Park Place, Cardiff CF10 3AT, Wales, UK
[3]Diamond Light Source, Harwell Science and Innovation Campus, Didcot OX11 0DE, UK
[4]Department of Structural Biology, Weizmann Institute of Science, Rehovot 7610001, Israel
[5]CNRS, Université de Bordeaux, ISM UMR 5255, 351 cours de la Libération, 33405 Talence Cedex, France

FG, 0000-0002-8900-2410; KDMH, 0000-0001-7855-8598

In a recent paper (Couzi *et al.* 2018 *R. Soc. open sci.* **5**, 180058. (doi:10.1098/rsos.180058)), we proposed a new phenomenological model to account for the I↔II↔"III" phase sequence in incommensurate *n*-alkane/urea inclusion compounds, which represents an alternative interpretation to that proposed in work of Toudic *et al.* In a Comment (Toudic *et al.* 2019 *R. Soc. open sci.* **6**, 182073. (doi:10.1098/rsos.182073)), Toudic *et al.* have questioned our assignment of the superspace group of phase II of *n*-nonadecane/urea, which they have previously assigned, based on a (3 + 2)-dimensional superspace, as $C222_1(00\gamma)(10\delta)$. In this Reply, we present new results from a comprehensive synchrotron single-crystal X-ray diffraction study of *n*-nonadecane/urea, involving measurements as a detailed function of temperature across the I↔II↔"III" phase transition sequence. Our results demonstrate conclusively that "main reflections" $(h, k, l, 0)$ with $h+k$ odd are observed in phase II of

$n$-nonadecane/urea (including temperatures in phase II that are just below the transition from phase I to phase II), in full support of our assignment of the (3+1)-dimensional superspace group $P2_12_12_1(00\gamma)$ to phase II. As our phenomenological model is based on phase II and phase 'III' of this incommensurate material having the _same_ (3+1)-dimensional superspace group $P2_12_12_1(00\gamma)$, it follows that the new X-ray diffraction results are in full support of our phenomenological model.

Our recently published paper [1] proposed a new phenomenological model, based on symmetry considerations and developed in the frame of a pseudospin-phonon coupling mechanism, to account for the mechanisms responsible for the I ↔ II ↔ 'III' phase sequence in the incommensurate $n$-hexadecane/ urea and $n$-nonadecane/urea inclusion compounds. This model is an alternative interpretation to that proposed by Toudic *et al.* in the work cited in their Comment [2].

Focusing on the case of $n$-nonadecane/urea, the model that we developed [1] is based on the use of $(3+1)$-dimensional superspace group descriptions for this incommensurate material, specifically $P6_122(00\gamma)$ in phase I and $P2_12_12_1(00\gamma)$ in both phase II and phase 'III'. By contrast, the work of Toudic *et al.* is based on the use of $(3+2)$-dimensional superspace groups for phases II and 'III'. Indeed, since a paper published in 2011 [3], Toudic and co-workers have discussed the I ↔ II ↔ 'III' phase sequences in $n$-nonadecane/urea extensively on this basis, assigning the $(3+2)$-dimensional superspace groups as $C222_1(00\gamma)(10\delta)$ for phase II and $P2_12_12_1(00\gamma)(00\delta)$ for phase 'III'. However, as discussed previously [4], the two misfit parameters ($\gamma$ and $\delta$) for $n$-nonadecane/urea invoked in the $(3+2)$-dimensional superspace group description of Toudic *et al.* are actually related by $\delta = -2 + 5\gamma$. This relationship between the two misfit parameters suggests that phases II and 'III' are more appropriately described by $(3+1)$-dimensional superspace groups as the minimal basis (with a single misfit parameter $\gamma$). In addition, our new phenomenological model [1] demonstrates that the thermal anomaly observed at the II ↔ 'III' phase transition may be attributed to a 'crossover' between two competing order parameters within the same phase II, described by superspace group $P2_12_12_1(00\gamma)$, without involving any symmetry breaking to a new phase 'III'.

Similarly, in the case of $n$-hexadecane/urea, Toudic *et al.* [5] describe phase 'III' of this material using a $(3+2)$-dimensional superspace group. However, it has also been shown [4] in this case that the two misfit parameters ($\gamma$ and $\delta$) are related by a simple relationship ($\delta = 2 - 4\gamma$), which again allows the symmetry properties to be described by a $(3+1)$-dimensional superspace group.

In their Comment [2], Toudic *et al.* focus on one point, specifically the claim that our previously published [4] single-crystal X-ray diffraction study on $n$-nonadecane/urea 'cannot be used to discuss the sequence of phases in this compound' because of the contention that the reported data at 147 K are too close in temperature to the so-called II ↔ 'III' transition, from which they suggest that our data from phase II are actually from phase 'III'. As Toudic *et al.* [2] suggest that our data reported previously [4] for phase II were (in their words) 'on the wrong phase', they make the further claim that our 'phenomenological description of the phase behaviour in $n$-nonadecane/urea is contrary to reliable experimental measurements'. In this Reply, we respond to the issues raised in the Comment of Toudic *et al.* [2].

The phase transition temperatures in $n$-nonadecane/urea have been reported previously [6] from thermal analysis techniques. From adiabatic calorimetry, the temperature of the I ↔ II phase transition is $(158.8 \pm 0.1)$ K and the temperature of the II ↔ 'III' phase transition is $(147.0 \pm 0.1)$ K. From differential scanning calorimetry (DSC) data recorded on cooling at 10 K min$^{-1}$, the corresponding temperatures are reported [6] to be $(157 \pm 1)$ K and $(140 \pm 1)$ K, respectively. In our previous single-crystal X-ray diffraction study of $n$-nonadecane/urea [4], the temperature (147 K) for study in phase II was selected on the basis of the reported phase transition temperatures from DSC data [6], as this temperature is close to the centre of the range representing phase II. However, the phase transition temperatures determined [6] from adiabatic calorimetry actually represent a more reliable indication of the phase transition temperatures in the context of the cooling schedule used in the single-crystal X-ray diffraction study. As the temperature of the II ↔ 'III' phase transition established from adiabatic calorimetry is $(147.0 \pm 0.1)$ K and as our reported single-crystal X-ray diffraction study for phase II was carried out at 147 K [4], there is uncertainty (depending, for example, on the accuracy of temperature control in the experimental measurement and on the reliability of the data on phase transition temperatures reported in [6]) whether the sample was actually in phase II or phase 'III' during the measurement at 147 K. This is the specific point that Toudic *et al.* have focused upon in their recent Comment [2]. In addition, Toudic *et al.* published an earlier Comment [7] on our previous X-ray diffraction study [4], in which they focused on exactly the same point as in their new Comment [2].

Instead of publishing a reply to their earlier Comment [7], we felt that the most satisfactory approach to address the issue would be to undertake a new and more comprehensive synchrotron single-crystal X-ray diffraction study, involving measurements of X-ray diffraction data for $n$-nonadecane/urea as a detailed function of temperature across the temperature range encompassing the I ↔ II ↔ 'III' phase transitions, in particular allowing clarification of the structural properties of phase II and phase 'III'. The results of our new study, which are in preparation for publication as a full paper [8], confirm the existence of 'main reflections' $(h, k, l, 0)$ with $h + k$ odd in phase II and in phase 'III', and are in full support of the structural conclusions reported in our earlier paper [4] concerning the assignment of the same $(3 + 1)$-dimensional superspace group $P2_12_12_1(00\gamma)$ to phase II and phase 'III' of $n$-nonadecane/urea. Importantly, our new results confirm that such reflections are observed at temperatures in phase II that are just below the phase transition from phase I to phase II, and therefore indisputably in phase II. Our new results are therefore in full support of our phenomenological model reported in [1]. However, given the uncertainty of whether the measurement of data at 147 K in our earlier paper [4] actually represented phase II or phase 'III', we accept that our previous statement [1] that 'it has been shown that both phases II and 'III' belong to the same superspace group $P2_12_12_1(00\gamma)$' (with [4] cited as evidence in support of this statement) was misleading, as it was not supported by the published evidence at that time. Nevertheless, we emphasize that any model is based on certain underlying assumptions, and one assumption in the case of our phenomenological model developed in [1] was that phase II and phase 'III' have the same superspace group $P2_12_12_1(00\gamma)$. Clearly this assumption, while not proven by published work at the time of publication of [1], is now supported by the new data presented here.

Our new synchrotron single-crystal X-ray diffraction study, carried out on beamline I19 at Diamond Light Source, involved the measurement of full data collections (following the data collection method described previously [4]) for a single crystal of $n$-nonadecane/urea (dimensions: $110 \times 40 \times 60\ \mu m^3$) at several temperatures on cooling in the following sequence: 300 K, 163 K to 145 K in increments of 2 K, and 145 K to 125 K in increments of 5 K. From 163 K to 125 K, cooling of the crystal between data collections was carried out at a rate of 1 K min$^{-1}$, and each new data collection was commenced only when the temperature was stable to within ±0.1 K of the target temperature. Importantly, by recording the single-crystal X-ray diffraction data at small $(\Delta T = 2\ K)$ temperature increments in the vicinity of the phase transition from phase I to phase II (and then at the same small temperature increments throughout phase II), the occurrence of the phase transition is verified by observing the change from hexagonal to orthorhombic symmetry in the X-ray diffraction data, rather than relying on the absolute value of the measured temperature alone. This protocol gives certainty in the measurement of X-ray diffraction data at temperatures that are indisputably in phase II (i.e. temperatures that are only a few K below the temperature at which the X-ray diffraction data indicate the occurrence of the phase transition from phase I to phase II).

From the measured X-ray diffraction data, reconstructed precession images were generated for several regions of reciprocal space of interest, and we focus here on the $(3, k, l)$ slice of reciprocal space as a function of temperature as it contains the $(3, -2, l)$ and $(3, 2, l)$ lines that were the focus of the discussion in our previous paper [4].

The X-ray diffraction data recorded at 300 K, 163 K and 161 K were indexed as hexagonal, indicating that the crystal is in phase I at these temperatures. The $(3, k, l)$ slice of reciprocal space at 161 K is shown in figure 1 (here the indices are specified relative to the orthohexagonal description of the hexagonal phase I in order to facilitate comparison to the results for phase II discussed below). We emphasize that, in figure 1, no diffraction maxima are observed for even values of $k$ (corresponding to $h + k$ odd) due to the $C$-centring of the hexagonal phase I when indexed using the orthohexagonal setting.

The X-ray diffraction data recorded at 159 K provide the first hint of the occurrence of the phase transition from phase I to phase II, with the data indexed as orthorhombic and with the appearance of some weak scattering along lines $(h, k, l)$ in reciprocal space with $h + k$ odd, as evident from the $(3, 2, l)$ and $(3, -2, l)$ lines in the $(3, k, l)$ slice shown in figure 2.

To discuss in more detail the structural behaviour of phase II, we focus on the data recorded at 157 K, for which the $(3, k, l)$ slice of reciprocal space is shown in figure 3. First of all, we emphasize that our measurement at 157 K indisputably represents phase II, as the measurement temperature (157 K) was only 4 K below a temperature (161 K) at which the material was clearly still in phase I, as discussed above. As the reported [6] temperature interval of phase II is 11.8 K (i.e. phase II exists across a temperature range of 11.8 K below the phase transition from phase I to phase II), our measurement at 157 K is therefore indisputably in phase II. Focusing on the $(3, 2, l)$ and $(3, -2, l)$ lines in figure 3, it is very clear that 'main reflections' are observed on these lines with indexing, in $(3 + 1)$-dimensional superspace $(h, k, l, m)$, corresponding to $l$ = integer and $m = 0$. Prominent examples of such reflections

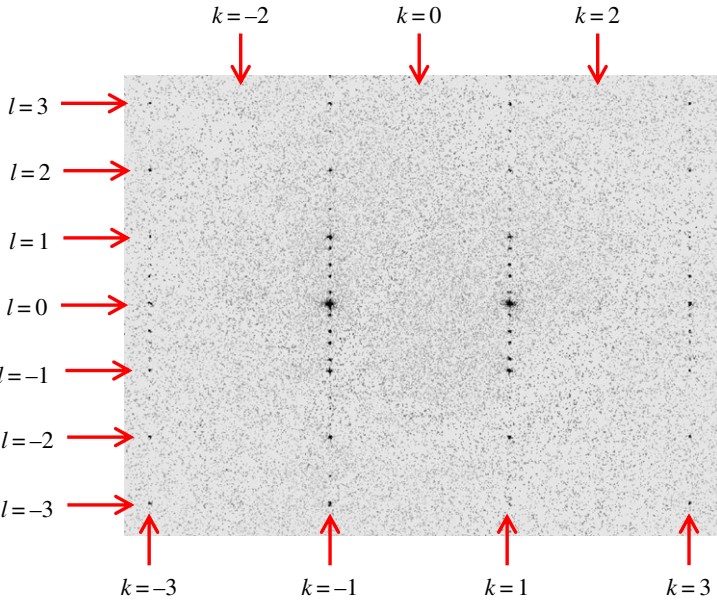

**Figure 1.** The (3, $k$, $l$) slice of reciprocal space recorded at 161 K (phase I).

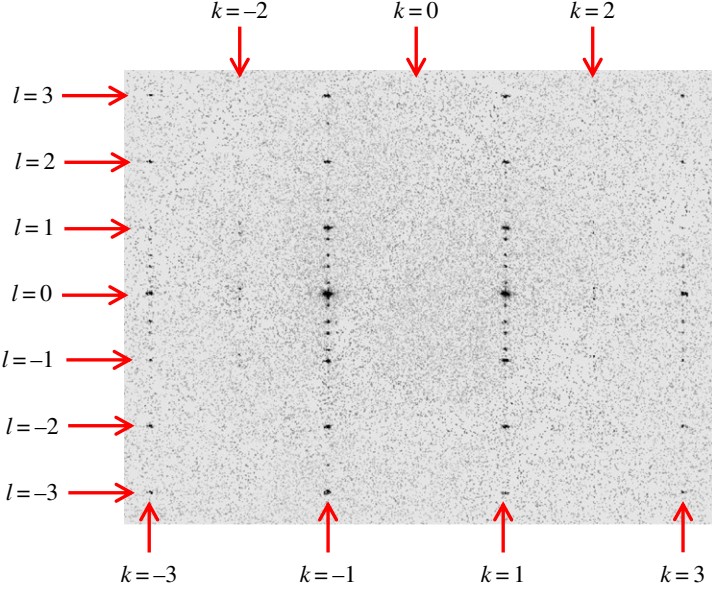

**Figure 2.** The (3, $k$, $l$) slice of reciprocal space recorded at 159 K, showing evidence for weak scattering in the (3, 2, $l$) and (3, −2, $l$) lines.

are (3, −2, 3, 0), (3, −2, 1, 0), (3, −2, 0, 0), (3, −2, −1, 0) and (3, −2, −3, 0) on the (3, −2, $l$) line, and (3, 2, 3, 0), (3, 2, 1, 0), (3, 2, 0, 0), (3, 2, −1, 0) and (3, 2, −3, 0) on the (3, 2, $l$) line. These observations are contradictory to the assertion of Toudic *et al.* that 'main reflections' ($h$, $k$, $l$, 0) with $h + k$ odd are absent in phase II.

Our data recorded at 155 K [which is also clearly in phase II as this temperature is only 6 K below a temperature (161 K) at which the sample was still in phase I] are also fully consistent with our conclusion that 'main reflections' of the type ($h$, $k$, $l$, 0) with $h + k$ odd are observed in phase II. At 155 K, data collections were carried out for two different attenuations of the incident beam, and the data shown in figure 4 were recorded at lower attenuation (representing an incident beam of higher intensity) than the data shown in figures 1–3. Figure 5 shows a zoomed view of the (3, −2, $l$) line at 155 K, from which it is incontrovertibly clear that the 'main reflections' indexed as (3, −2, 1, 0), (3, −2, 0, 0) and (3, −2, −1, 0) in the (3 + 1)-dimensional superspace are present in phase II. The horizontal blue lines in figure 5 confirm that

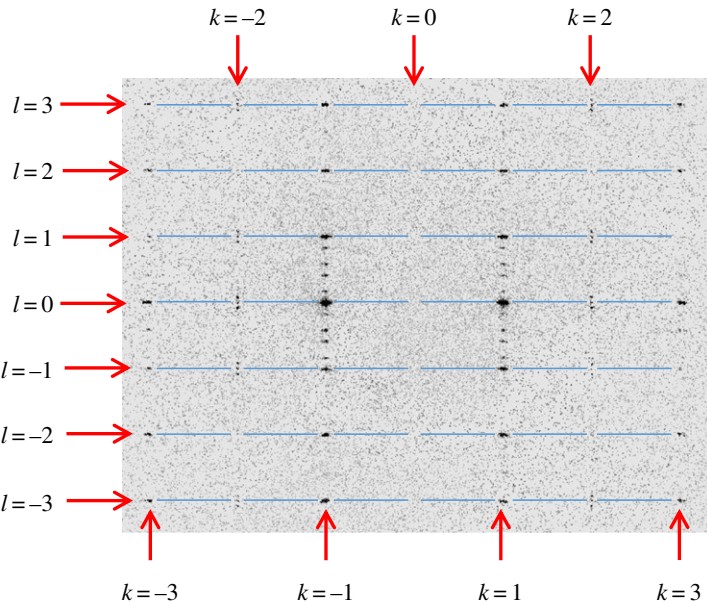

**Figure 3.** The $(3, k, l)$ slice of reciprocal space recorded for phase II at 157 K, showing clearly the presence of reflections in the $(3, 2, l)$ and $(3, -2, l)$ lines for the orthorhombic phase II. Importantly, 'main reflections' are observed on these lines with indexing, in $(3 + 1)$-dimensional superspace $(h, k, l, m)$, corresponding to $l$ = integer and $m = 0$. The other reflections observed on the $(3, 2, l)$ and $(3, -2, l)$ lines are 'satellite reflections' with $m \neq 0$. The horizontal blue lines are added to verify that the 'main reflections' on the $(3, 2, l)$ and $(3, -2, l)$ lines (which have $h + k$ odd) occur at exactly the same integer values of $l$ (with $m = 0$) as the 'main reflections' on the $(3, 3, l)$, $(3, 1, l)$, $(3, -1, l)$ and $(3, -3, l)$ lines (which have $h + k$ even).

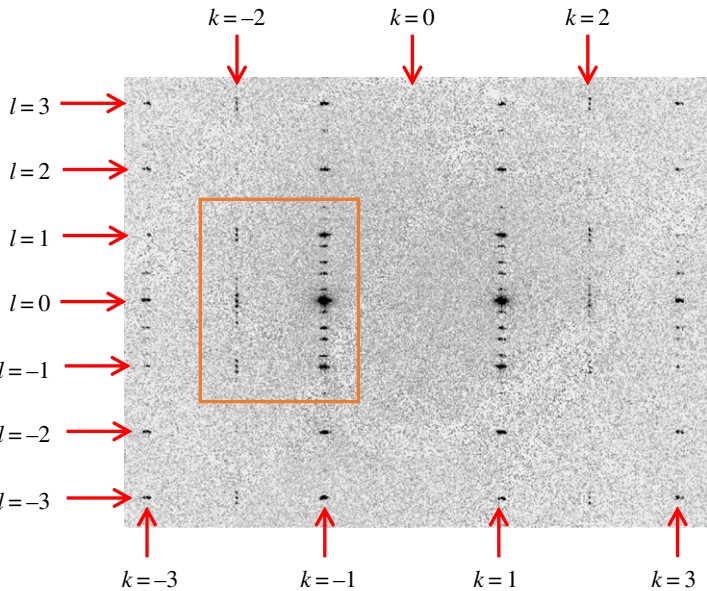

**Figure 4.** The $(3, k, l)$ slice of reciprocal space recorded for phase II at 155 K (figure 3), showing clearly the presence of reflections in the $(3, 2, l)$ and $(3, -2, l)$ lines (which have $h + k$ odd). As at 157 K (figure 3), 'main reflections' are clearly observed on these lines with indexing, in $(3 + 1)$-dimensional superspace $(h, k, l, m)$, corresponding to $l$ = integer and $m = 0$, together with 'satellite reflections' corresponding to $m \neq 0$. The region within the orange rectangle is expanded in figure 5.

these 'main reflections' on the $(3, -2, l)$ line (with $h + k$ odd) have the same exact integer values of $l$ (with $m = 0$) as the 'main reflections' on the $(3, -1, l)$ line (with $h + k$ even), which is also shown. We note that the other reflections observed along the $(3, -2, l)$ line are 'satellite reflections' $(3, -2, l, m)$ with $m \neq 0$, and the indexing $(l, m)$ of these 'satellite reflections' corresponds to the indexing given in fig. 2a of [4].

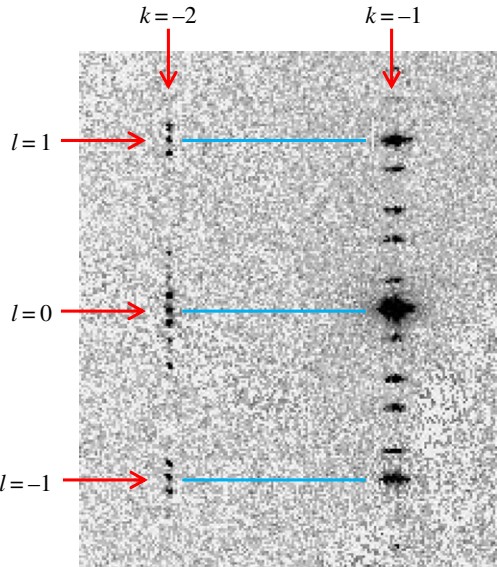

**Figure 5.** Zoomed region of the (3, $k$, $l$) slice of reciprocal space recorded for phase II at 155 K (from the same data as figure 4), verifying that, on the (3, −2, $l$) line, 'main reflections' ($h$, $k$, $l$, $m$) with $l$ = integer and $m$ = 0 corresponding to (3, −2, 1, 0), (3, −2, 0, 0) and (3, −2, −1, 0) are observed, and occur at exactly the same integer values of $l$ as the 'main reflections' on the (3, −1, $l$) line.

While we have focused here on the (3, $k$, $l$) slice of reciprocal space, further evidence that 'main reflections' of the type ($h$, $k$, $l$, 0) with $h + k$ odd are observed in phase II is seen in many other regions of reciprocal space, and a more comprehensive discussion will be presented in a subsequent full paper [8].

Our new data also confirm that 'main reflections' of the type ($h$, $k$, $l$, 0) with $h + k$ odd are observed at 153 K and at all lower temperatures studied (down to 125 K), which includes temperatures corresponding to phase 'III'. The presence of such reflections in phase 'III' was also reported in our previous paper [4] (fig. 3a of [4]) and is in agreement with the interpretations of phase 'III' of Toudic *et al.* (although, as noted above, Toudic *et al.* describe phase 'III' using a (3 + 2)-dimensional superspace, while it has been shown [4] that a (3 + 1)-dimensional superspace description is sufficient as the minimal basis).

In conclusion, from our new synchrotron single-crystal X-ray diffraction data, it is indisputable that 'main reflections' ($h$, $k$, $l$, 0) with $h + k$ odd are observed in phase II of $n$-nonadecane/urea, which is in full support of our assignment of the (3 + 1)-dimensional superspace group $P2_12_12_1(00\gamma)$ to phase II. As our phenomenological model [1] for rationalization of the phase transitions in this incommensurate material is based on phase II and phase 'III' having the _same_ (3 + 1)-dimensional superspace group $P2_12_12_1(00\gamma)$, it follows that the results from our new X-ray diffraction study are in full support of our phenomenological model [1], which represents a valid basis for understanding the phase transition mechanisms in incommensurate $n$-alkane/urea inclusion compounds.

Finally, we emphasize that an important question is now to understand the reasons underlying the differences between the experimental diffraction data, and the corresponding structural conclusions, for $n$-nonadecane/urea reported in the work of Toudic *et al.* (see fig. 2b of [3]) and in our own work (as shown by the data reported here and in our future full paper [8]), particularly concerning the structural interpretation of phase II.

Data accessibility. This article does not contain any additional data.

Authors' contributions. The authors (M.C., F.G. and K.D.M.H.) of the phenomenological model reported in [1] contributed equally to preparing this Reply, which includes preliminary results from a new synchrotron single-crystal X-ray diffraction study that will be published in full in a future paper [8]. The new X-ray diffraction experiments were carried out on beamline I19 at Diamond Light Source by K.C., R.P. and K.D.M.H. using a sample prepared by P.A.W. The data analysis was carried out by K.C., and all authors participated in the discussion and interpretation of the results. All authors have given their final approval for publication.

Competing interests. We declare we have no competing interests.

Funding. We received no funding for this study.

Acknowledgements. We are grateful to Diamond Light Source for the award of experimental time on beamline I19 (proposal MT21340), and for the award of a PhD studentship (to R.P.).

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
