## [Reviewer comments · Royal Society Open Science]

Review History

RSOS-190518.R0 (Original submission)

Review form: Reviewer 1

Is the manuscript scientifically sound in its present form?

Yes

Are the interpretations and conclusions justified by the results?

No

Is the language acceptable?

Yes

Is it clear how to access all supporting data?

Not Applicable

Do you have any ethical concerns with this paper?

No

Have you any concerns about statistical analyses in this paper?

No

Recommendation?

Major revision is needed (please make suggestions in comments)

Comments to the Author(s)

It is shown in Table 1 in Toudic et al 2019 comment that some diffraction peaks for "Phase III" are not present for "Phase II", indicating different structures. So how can these two structures still be the same phase? Please explain.

Review form: Reviewer 2

Is the manuscript scientifically sound in its present form?

Yes

Are the interpretations and conclusions justified by the results?

Yes

Is the language acceptable?

Yes

Is it clear how to access all supporting data?

Not Applicable

Do you have any ethical concerns with this paper?

No

Have you any concerns about statistical analyses in this paper?

No

Recommendation?

Accept as is

Comments to the Author(s)

Hopefully, we can see the data in their mentioned reference 6.

Review form: Reviewer 3

Is the manuscript scientifically sound in its present form?

No

Are the interpretations and conclusions justified by the results?

No

Is the language acceptable?

No

Is it clear how to access all supporting data?

No

Do you have any ethical concerns with this paper?

Yes

Have you any concerns about statistical analyses in this paper?

No

Recommendation?

Major revision is needed (please make suggestions in comments)

Comments to the Author(s)

I refer to the two "sides" in this debate as "Toudic" and "Couzi". References are the numbers used in this submitted comment; I'll call this reference [0].

This paper is a response to a comment by Toudic et al [2] on work that Couzi et al published in RSOS in 2018 [1]. I have seen the comment by Toudic [2], so review accordingly.

It seems that a significant disagreement has arisen between Toudic and Couzi concerning the complex incommensurate phase transitions in a family of urea inclusion compounds. I have done my best to understand the situation based on the various publications in question. It is, however, hard to reach a definitive conclusion as the arguments involved are complex both scientifically and non-scientifically. My conclusions are based on my best understanding of the situation, and I apologise if I have reached incorrect conclusions.

The crux of the scientific argument seems to be whether there is a single transition from phase I to phase II in these materials or a second lower-temperature transition to phase III. This has, however, spilled over into a debate about whether the scientific literature is being appropriately interpreted and cited.

The scientific facts rest on the extinction patterns seen in phase II. In a 2016 EPL article [4] Couzi et al stated that only phase II was real. This was challenged by Toudic [5] who stated that the temperature used in [4] meant that Couzi were actually studying phase III. This comment was not challenged by Couzi.

In the paper currently under question [1] (published after [5]) Couzi et al. make some strong statements which, to my reading, don't take the content of [5] appropriately into account. In particular they say (P4) "[our EPL] recent paper...demonstrated conclusively that the symmetries of both phase II and "III" can be described by (3+1)-dimensional superspace groups" (and similar on P5). In my view it is remiss that they did not explicitly refer to the controversy over this point that was discussed in [5]. The fact that Couzi was fully aware of the controversy is supported by paragraph 4 of this comment [0] which says that the only way to resolve things was to collect more data. There are also other phrases (e.g. "...admitting explicitly...." P5) in [1] which seem overly-strongly worded based on what I understand of the literature.

In this response [0] Couzi is now essentially saying that he has definitive proof for the crucial h+k reflections being present in phase II. If true this would clear up the scientific controversy. Unfortunately this is just "to be published [6]" and there is no evidence presented with this

comment to support the statement. As such this comment doesn't follow the usual process of scientific peer review. The reviewer and reader are being asked to accept a statement in a controversial area without being given the normal opportunity to review the experimental data. This seems to go against the "Open" ethos or RSOS. I imagine Toudic will feel the same way.

My impression is that publication of these back-to back comments as written will lead to future arguments and both professional and personal upset for all involved. It will also create a confused literature for those that follow. These are things things RSOS should try and avoid.

The best route forward is obviously something that the editorial team must make the final decision on, as there as many policy as science issues in play. However, my recommendation is that this response should only be published if it is toned down and Couzi is more conciliatory in the language used. I suggest Couzi acknowledges that their statements in [1] were too strong in places based on the data then publically available. Any additional statements they want to include in this comment about subsequent work have to be supported with experimental data that others can judge (e.g. as SI or via link to a preprint of any future publication such as [6]).

Decision letter (RSOS-190518.R0)

02-May-2019

Dear Dr Guillaume:

Title: Reply to Comment on Couzi et al. (2018): A phenomenological model for structural phase transitions in incommensurate alkane/urea inclusion compounds
Manuscript ID: RSOS-190518

The editor assigned to your manuscript has now received comments from reviewers. We would like you to revise your paper in accordance with the referee and Subject Editor suggestions which can be found below (not including confidential reports to the Editor). Please note this decision does not guarantee eventual acceptance.

Please submit your revised paper before 25-May-2019. Please note that the revision deadline will expire at 00.00am on this date. If we do not hear from you within this time then it will be assumed that the paper has been withdrawn. In exceptional circumstances, extensions may be possible if agreed with the Editorial Office in advance. We do not allow multiple rounds of revision so we urge you to make every effort to fully address all of the comments at this stage. If deemed necessary by the Editors, your manuscript will be sent back to one or more of the original reviewers for assessment. If the original reviewers are not available we may invite new reviewers.

When submitting your revised manuscript, you must respond to the comments made by the referees and upload a file "Response to Referees" in "Section 6 - File Upload". Please use this to

document how you have responded to the comments, and the adjustments you have made. In order to expedite the processing of the revised manuscript, please be as specific as possible in your response.

Please also include the following statements alongside the other end statements. As we cannot publish your manuscript without these end statements included, if you feel that a given heading is not relevant to your paper, please nevertheless include the heading and explicitly state that it is not relevant to your work.

- Ethics statement

Please clarify whether you received ethical approval from a local ethics committee to carry out your study. If so please include details of this, including the name of the committee that gave consent in a Research Ethics section after your main text. Please also clarify whether you received informed consent for the participants to participate in the study and state this in your Research Ethics section.

OR

Please clarify whether you obtained the necessary licences and approvals from your institutional animal ethics committee before conducting your research. Please provide details of these licences and approvals in an Animal Ethics section after your main text.

OR

Please clarify whether you obtained the appropriate permissions and licences to conduct the fieldwork detailed in your study. Please provide details of these in your methods section.

- Data accessibility

It is a condition of publication that you make available the data and research materials supporting the results in the article. Datasets should be deposited in an appropriate publicly available repository and details of the associated accession number, link or DOI to the datasets must be included in the Data Accessibility section of the article (<http://royalsocietypublishing.org/instructions-authors#question17>). Reference(s) to datasets should also be included in the reference list of the article with DOIs (where available).

Please include a Data Availability section after your main text stating where supporting data are available from, or where they will be made available should your article be accepted for publication.

If you wish to submit your supporting data or code to Dryad (<http://datadryad.org/>), or modify your current submission to dryad, please use the following link:
<http://datadryad.org/submit?journalID=RSOS&manu=RSOS-190518>

- Competing interests

Please include a Competing Interests section after your main text declaring any financial or non-financial competing interests. If you have no competing interests please state 'I/we have no competing interests.'

- Authors' contributions

Please include an Authors' Contributions section at the end of your main text detailing the contribution of each author. All authors should have read and approved the manuscript before submission and this should be stated in the Authors' Contributions section.

The list of Authors should meet all of the following criteria; 1) substantial contributions to conception and design, or acquisition of data, or analysis and interpretation of data; 2) drafting the article or revising it critically for important intellectual content; and 3) final approval of the version to be published.

- Acknowledgements

- Funding statement

Please include a funding section after your main text which lists the source of funding for each author.

On behalf of the Subject Editor Professor Anthony Stace.

RSC Associate Editor: 1

Comments to the Author:

A number of reviewers have commented on this Reply to a peer-reviewed Comment. It seems that a number of matters need to be resolved in the Reply before it can be considered for publication - most notably, the request for data to be made available. Please respond accordingly.

Reviewers' Comments to Author:

Reviewer: 1

Comments to the Author(s)

It is shown in Table 1 in Toudic et al 2019 comment that some diffraction peaks for "Phase III" are not present for "Phase II", indicating different structures. So how can these two structures still be the same phase? Please explain.

Reviewer: 2

Comments to the Author(s)

Hopefully, we can see the data in their mentioned reference 6.

Reviewer: 3

Comments to the Author(s)

I refer to the two "sides" in this debate as "Toudic" and "Couzi". References are the numbers used in this submitted comment; I'll call this reference [0].

This paper is a response to a comment by Toudic et al [2] on work that Couzi et al published in RSOS in 2018 [1]. I have seen the comment by Toudic [2], so review accordingly.

It seems that a significant disagreement has arisen between Toudic and Couzi concerning the complex incommensurate phase transitions in a family of urea inclusion compounds. I have done my best to understand the situation based on the various publications in question. It is, however, hard to reach a definitive conclusion as the arguments involved are complex both scientifically and non-scientifically. My conclusions are based on my best understanding of the situation, and I apologise if I have reached incorrect conclusions.

The crux of the scientific argument seems to be whether there is a single transition from phase I to phase II in these materials or a second lower-temperature transition to phase III. This has, however, spilled over into a debate about whether the scientific literature is being appropriately interpreted and cited.

The scientific facts rest on the extinction patterns seen in phase II. In a 2016 EPL article [4] Couzi et al stated that only phase II was real. This was challenged by Toudic [5] who stated that the temperature used in [4] meant that Couzi were actually studying phase III. This comment was not challenged by Couzi.

In the paper currently under question [1] (published after [5]) Couzi et al. make some strong statements which, to my reading, don't take the content of [5] appropriately into account. In particular they say (P4) "[our EPL] recent paper...demonstrated conclusively that the symmetries of both phase II and "III" can be described by (3+1)-dimensional superspace groups" (and similar on P5). In my view it is remiss that they did not explicitly refer to the controversy over this point that was discussed in [5]. The fact that Couzi was fully aware of the controversy is supported by paragraph 4 of this comment [0] which says that the only way to resolve things was to collect more data. There are also other phrases (e.g. "...admitting explicitly...." P5) in [1] which seem overly-strongly worded based on what I understand of the literature.

In this response [0] Couzi is now essentially saying that he has definitive proof for the crucial $h+k$ reflections being present in phase II. If true this would clear up the scientific controversy. Unfortunately this is just "to be published [6]" and there is no evidence presented with this comment to support the statement. As such this comment doesn't follow the usual process of scientific peer review. The reviewer and reader are being asked to accept a statement in a controversial area without being given the normal opportunity to review the experimental data. This seems to go against the "Open" ethos or RSOS. I imagine Toudic will feel the same way.

The best route forward is obviously something that the editorial team must make the final decision on, as there are many policy issues in play. However, my recommendation is that this response should only be published if it is toned down and Couzi is more conciliatory in

the language used. I suggest Couzi acknowledges that their statements in [1] were too strong in places based on the data then publically available. Any additional statements they want to include in this comment about subsequent work have to be supported with experimental data that others can judge (e.g. as SI or via link to a preprint of any future publication such as [6]).

Author's Response to Decision Letter for (RSOS-190518.R0)

See Appendix A.

RSOS-190518.R1 (Revision)

Review form: Reviewer 1

Is the manuscript scientifically sound in its present form?

Yes

Are the interpretations and conclusions justified by the results?

Yes

Is the language acceptable?

Yes

Is it clear how to access all supporting data?

Yes

Do you have any ethical concerns with this paper?

No

Have you any concerns about statistical analyses in this paper?

No

Recommendation?

Accept as is

Comments to the Author(s)

The authors addressed the question I raised and I greatly appreciate it.

Review form: Reviewer 2

Is the manuscript scientifically sound in its present form?

Yes

Are the interpretations and conclusions justified by the results?

Yes

Is the language acceptable?

Yes

Do you have any ethical concerns with this paper?

No

Recommendation?

Accept as is

Comments to the Author(s)

The new proofs sound reasonable to me.

Review form: Reviewer 3

Is the manuscript scientifically sound in its present form?

Yes

Are the interpretations and conclusions justified by the results?

Yes

Is the language acceptable?

No

Is it clear how to access all supporting data?

Not Applicable

Do you have any ethical concerns with this paper?

Yes

Have you any concerns about statistical analyses in this paper?

No

Recommendation?

Accept with minor revision (please list in comments)

Comments to the Author(s)

I have read the modified comment and the responses of the authors. The authors have provided new experimental data for one of the compounds under debate as requested by the referees, and the data show that their Phase II has the same extinction conditions as reported for Phase III by Toudic et al. As such the new data presented addresses the core question posed in the Toudic comment [2]. The authors have included an acknowledgement that some of the statements in [1] weren't proven at the time of writing which seems appropriate.

My recommendation is to publish this response but I have the following suggestion:

P3 L4-6: I don't think that the sentence "Toudic et al do not contest our assertion that the

diffraction data for both phase II and phase "III" can be indexed ... (3+1) dimensional superspace" should be included in this Reply. It addresses an issue that wasn't raised in the RSOS comment. The issue was, however, discussed extensively in the earlier EPL comment by Toudic [7] to which Couzi didn't respond. The debate here also concerns the n-hexadecane compound, and no new data is presented on this. P6 L37 should be updated at the same time. Any further discussion on this point should be via regular peer-reviewed publication.

P3 L37: I would suggest "vindicated" is changed to "supported". Vindicated, at least in its etymology, can be interpreted as a somewhat aggressive term. As such it doesn't mirror the acknowledgement in the preceding section that statements were made without scientific evidence.

Decision letter (RSOS-190518.R1)

24-Jun-2019

Dear Dr Guillaume:

Title: Reply to Comment on Couzi et al. (2018): A phenomenological model for structural phase transitions in incommensurate alkane/urea inclusion compounds
Manuscript ID: RSOS-190518.R1

Thank you for submitting the above manuscript to Royal Society Open Science. On behalf of the Editors and the Royal Society of Chemistry, I am pleased to inform you that your manuscript will be accepted for publication in Royal Society Open Science subject to minor revision in accordance with the referee suggestions. Please find the reviewers' comments at the end of this email.

The reviewers and handling editors have recommended publication, but also suggest some minor revisions to your manuscript. Therefore, I invite you to respond to the comments and revise your manuscript.

Because the schedule for publication is very tight, it is a condition of publication that you submit the revised version of your manuscript before 03-Jul-2019. Please note that the revision deadline will expire at 00.00am on this date. If you do not think you will be able to meet this date please let me know immediately.

Best wishes,
Dr Laura Smith
Publishing Editor, Journals

RSC Associate Editor:
Comments to the Author:
Please make the required changes to the text as recommended by Reviewer 3.

RSC Subject Editor:
Comments to the Author:
(There are no comments.)

Reviewer comments to Author:

Reviewer: 1

Comments to the Author(s)

The authors addressed the question I raised and I greatly appreciate it.

Reviewer: 2

Comments to the Author(s)

The new proofs sound reasonable to me.

Reviewer: 3

Comments to the Author(s)

I have read the modified comment and the responses of the authors. The authors have provided new experimental data for one of the compounds under debate as requested by the referees, and the data show that their Phase II has the same extinction conditions as reported for Phase III by Toudic et al. As such the new data presented addresses the core question posed in the Toudic comment [2]. The authors have included an acknowledgement that some of the statements in [1] weren't proven at the time of writing which seems appropriate.

My recommendation is to publish this response but I have the following suggestion:

P3 L4-6: I don't think that the sentence "Toudic et al do not contest our assertion that the diffraction data for both phase II and phase "III" can be indexed ... (3+1) dimensional superspace" should be included in this Reply. It addresses an issue that wasn't raised in the RSOS comment. The issue was, however, discussed extensively in the earlier EPL comment by Toudic [7] to which Couzi didn't respond. The debate here also concerns the n-hexadecane compound, and no new data is presented on this. P6 L37 should be updated at the same time. Any further discussion on this point should be via regular peer-reviewed publication.

P3 L37: I would suggest "vindicated" is changed to "supported". Vindicated, at least in its etymology, can be interpreted as a somewhat aggressive term. As such it doesn't mirror the acknowledgement in the preceding section that statements were made without scientific evidence.

Author's Response to Decision Letter for (RSOS-190518.R1)

See Appendix B.

RSOS-190518.R2 (Revision)

Review form: Reviewer 3

Is the manuscript scientifically sound in its present form?

Yes

Are the interpretations and conclusions justified by the results?

Yes

Is the language acceptable?

Yes

Do you have any ethical concerns with this paper?

No

Have you any concerns about statistical analyses in this paper?

No

Recommendation?

Accept as is

Comments to the Author(s)

I am happy that the authors have accepted the recommendations made.

Decision letter (RSOS-190518.R2)

22-Jul-2019

Dear Dr Guillaume:

Title: Reply to Comment on Couzi et al. (2018): A phenomenological model for structural phase transitions in incommensurate alkane/urea inclusion compounds

Manuscript ID: RSOS-190518.R2

It is a pleasure to accept your manuscript in its current form for publication in Royal Society Open Science. The chemistry content of Royal Society Open Science is published in collaboration with the Royal Society of Chemistry.

RSC Associate Editor:
Comments to the Author:
(There are no comments.)

RSC Subject Editor:
Comments to the Author:
(There are no comments.)

Reviewer(s)' Comments to Author:
Reviewer: 3

Comments to the Author(s)
I am happy that the authors have accepted the recommendations made.

Appendix A

The comments of the referees are in red, and our responses are in black.

Reviewer: 1

It is shown in Table 1 in Toudic et al 2019 comment that some diffraction peaks for "Phase III" are not present for "Phase II", indicating different structures. So how can these two structures still be the same phase? Please explain.

The concern of the Referee is addressed by the new data presented in our revised Reply, which clearly show that certain diffraction peaks that are claimed by Toudic et al to be absent in Phase II but present in Phase III, are actually observed in both Phase II and Phase III. The diffraction peaks in question are the "main reflections" $(h, k, l, 0)$ with $h+k$ odd. It is very clear from Figures 3, 4 and 5 of our revised Reply that several reflections of this type are indisputably present in Phase II. For example, the reflections $(3, -2, 1, 0)$, $(3, -2, 0, 0)$ and $(3, -2, -1, 0)$ are explicitly highlighted in Figure 5.

We note that, as the Comment of Toudic et al in 2019 does not contain a Table, we believe that the referee is actually referring to Table 1 in an earlier Comment by Toudic et al (ref. 7 of our revised Reply). The new data for phase II included in our revised Reply demonstrate that all diffraction peaks listed in Table 1 of Toudic et al 2019 (under the heading "Presence in Couzi data") are indeed present in our data for phase II. We emphasize that the new data included in the revised Reply were recorded at small temperature increments on crossing the phase transition from phase I to phase II such that we are certain that our measured data do indeed represent phase II.

As noted in the final sentence of our revised Reply, "an important question is now to understand the reasons underlying the differences between the experimental diffraction data, and the corresponding structural conclusions, for n -nonadecane/urea reported in the work of Toudic et al and in our own work". We are fully confident in the reliability of the new data and conclusions presented in our Reply, and we hope that progress will be made in the future to understand why the data reported previously by Toudic et al for phase II of n -nonadecane/urea do not agree with our results.

Reviewer: 2

Hopefully, we can see the data in their mentioned reference 6.

We perfectly understand that this point is crucial in our response to the Comment of Toudic et al, and we have now included our new data in the revised Reply.

As discussed in our revised Reply, the new data clearly show the existence of "main reflections" $(h, k, l, 0)$ with $h+k$ odd in phase II of n -nonadecane/urea, which is the main point of disagreement with the results of Toudic et al, who claim that such reflections are absent in phase II. Figures 3, 4 and 5 of our revised Reply show clearly the presence of several reflections of this type in Phase II. For example, the reflections $(3, -2, 1, 0)$, $(3, -2, 0, 0)$ and $(3, -2, -1, 0)$ are explicitly highlighted in Figure 5.

Furthermore, we emphasize that, by recording the new data at small temperature increments on crossing the phase transition from phase I to phase II, we are certain that our measurements are definitely in phase II. Thus, the data at 161 K are still the hexagonal high-temperature phase I, while the first evidence of the transition to the orthorhombic phase II is observed at 159 K, and the data at 157 K and 155 K indisputably represent phase II. As the temperature interval of phase II is 11.8 K (i.e.

phase II exists in a temperature interval of 11.8 K below the phase transition from phase I to phase II), our measurements at 155 K and 157 K are therefore definitely in phase II.

Reviewer: 3

I refer to the two "sides" in this debate as "Toudic" and "Couzi". References are the numbers used in this submitted comment; I'll call this reference [0].

This paper is a response to a comment by Toudic et al [2] on work that Couzi et al published in RSOS in 2018 [1]. I have seen the comment by Toudic [2], so review accordingly.

It seems that a significant disagreement has arisen between Toudic and Couzi concerning the complex incommensurate phase transitions in a family of urea inclusion compounds. I have done my best to understand the situation based on the various publications in question. It is, however, hard to reach a definitive conclusion as the arguments involved are complex both scientifically and non-scientifically. My conclusions are based on my best understanding of the situation, and I apologise if I have reached incorrect conclusions.

The crux of the scientific argument seems to be whether there is a single transition from phase I to phase II in these materials or a second lower-temperature transition to phase III. This has, however, spilled over into a debate about whether the scientific literature is being appropriately interpreted and cited.

The scientific facts rest on the extinction patterns seen in phase II. In a 2016 EPL article [4] Couzi et al stated that only phase II was real. This was challenged by Toudic [5] who stated that the temperature used in [4] meant that Couzi were actually studying phase III. This comment was not challenged by Couzi.

We agree completely with the reviewer that *"the scientific facts rest on the extinction patterns seen in phase II"*. In this regard, the new data presented in our revised Reply verify that "main reflections" ($h, k, l, 0$) with $h+k$ odd are clearly observed in phase II.

In our revised Reply, we have also added a discussion (see the paragraph beginning "The phase transition temperatures ..." on page 2 of the revised Reply) relating to the selection of 147 K as the temperature for our earlier X-ray diffraction study (i.e. ref. [4] of our revised Reply), and accepting that there is *"uncertainty whether the sample was actually in phase II or phase 'III' during the measurement at 147 K"*. We believe that Toudic is not correct (as quoted by the reviewer) to state that "the temperature used in [4] meant that Couzi were actually studying phase III", but rather that we could not be certain whether the sample was actually in phase II or in phase III at this temperature.

The fact that we did not "challenge" the earlier Comment of Toudic (i.e. ref. [7] of our revised Reply) does not mean that we agreed with the earlier Comment of Toudic, but instead (as stated in our Reply) "we felt that the most satisfactory approach to address the issue would be to undertake a new and more comprehensive synchrotron single-crystal X-ray diffraction study". In hindsight, we should probably have proceeded to publish a Reply to the earlier Comment of Toudic (as the absence of a Reply might have given the false impression that we agreed with their Comment). Nevertheless, we are optimistic that the new data presented here will resolve the controversy.

However, as noted in the final sentence of our revised Reply, *"an important question is now to understand the reasons underlying the differences between the experimental diffraction data, and the corresponding structural conclusions, for n-nonadecane/urea reported in the work of Toudic et al. and in our own work"*. We are fully confident in the reliability of the new data and conclusions presented in our Reply, and we hope that progress will be made in the future to understand why the data reported previously by Toudic et al for phase II of n-nonadecane/urea do not agree with our results.

In the paper currently under question [1] (published after [5]) Couzi et al. make some strong statements which, to my reading, don't take the content of [5] appropriately into account. In particular they say (P4) [our EPL] recent paper demonstrated conclusively that the symmetries of both phase II and can be described by (3+1)-dimensional superspace groups (and similar on P5). In my view it is remiss that they did not explicitly refer to the controversy over this point that was discussed in [5]. The fact that Couzi was fully aware of the controversy is supported by paragraph 4 of this comment [0] which says that the only way to resolve things was to collect more data. There are also other phrases (e.g. admitting explicitly P5) in [1] which seem overly-strongly worded based on what I understand of the literature.

Our previous statement (in ref. [1] of our revised Reply) that our EPL paper (ref. [4] of our revised Reply) "demonstrated conclusively that the symmetries of both phase II and can be described by (3+1)-dimensional superspace groups" is based on our proof that the two misfit parameters proposed previously by Toudic are actually related by a simple relationship, such that all diffraction maxima can be indexed on the basis of a single misfit parameter and hence indexed on the basis of a minimal (3+1)-dimensional superspace. This point was discussed in detail in our EPL paper, and this aspect of the EPL paper still holds firmly. The "controversy" only concerns whether the 147 K data reported in our EPL paper actually represented phase II or phase "III", with implications for the assignment of the superspace group for phase II.

Significantly, the recent Comment of Toudic (ref. [2] of our revised Reply) makes no criticism of our assertion that the minimal (3+1)-dimensional superspace descriptions are appropriate for this system. Instead, their criticism is focused solely on the uncertainty of whether the data at 147 K reported in our EPL paper represented phase II or phase "III". The new data presented in our revised Reply clarifies this "controversy", as the data recorded at temperatures that are indisputably in phase II verify that "main reflections" ($h, k, l, 0$) with $h+k$ odd are indeed present in phase II.

In this response [0] Couzi is now essentially saying that he has definitive proof for the crucial $h+k$ reflections being present in phase II. If true this would clear up the scientific controversy. Unfortunately this is just "to be published [6]" and there is no evidence presented with this comment to support the statement. As such this comment doesn't follow the usual process of scientific peer review. The reviewer and reader are being asked to accept a statement in a controversial area without being given the normal opportunity to review the experimental data. This seems to go against the "Open" ethos or RSOS. I imagine Toudic will feel the same way.

We totally understand the need to provide our new data to allow the reviewers to assess the evidence, and the revised version of our Reply has been substantially extended to include results from our new experiments. We apologize for not including this information at the time of submitting our original Reply.

We are glad that the reviewer agrees that the scientific controversy would be cleared up by providing definitive proof of the "crucial reflections" in phase II. In this regard, we emphasize (i) that the new data in the revised Reply were recorded at small temperature increments on crossing the phase transition from phase I to phase II such that we can be certain that our measured data do indeed represent phase II, and (ii) that the new data for phase II show clear evidence that "main reflections" $(h, k, l, 0)$ with $h+k$ odd are present in phase II. Both of these points are elaborated in detail in the revised Reply.

Firstly, we note that the data at 161 K are still the hexagonal high-temperature phase I, while the first evidence of the transition to the orthorhombic phase II is observed at 159 K, and the data at 157 K and 155 K indisputably represent phase II. As the temperature interval of phase II is 11.8 K (i.e. phase II exists in a temperature interval of 11.8 K below the phase transition from phase I to phase II), our measurements at 157 K and 155 K are indisputably in phase II.

Secondly, Figures 3, 4 and 5 of the revised Reply show clearly the presence of reflections of the type $(h, k, l, 0)$ with $h+k$ odd in phase II. For example, the reflections $(3, -2, 1, 0)$, $(3, -2, 0, 0)$ and $(3, -2, -1, 0)$ are explicitly highlighted in Figure 5.

The best route forward is obviously something that the editorial team must make the final decision on, as there as many policy as science issues in play. However, my recommendation is that this response should only be published if it is toned down and Couzi is more conciliatory in the language used. I suggest Couzi acknowledges that their statements in [1] were too strong in places based on the data then publically available.

We agree with the referee on this point, and we have looked carefully at how the statements in our RSOS paper (ref. [1] of our revised Reply) may have been based too strongly on those conclusions in our previous EPL paper (ref. [4] of our revised Reply) that relied upon the data recorded at 147 K in that paper. For this reason, we feel that the statement in our RSOS paper "*it has been shown that both phases II and "III" belong to the same superspace group $P2_12_12_1(00\gamma)$* " was too strongly worded, and we have added the following statement to our revised Reply in order to address this issue:

... given the uncertainty of whether the measurement of data at 147 K in our earlier paper [4] actually represented phase II or phase "III", we accept that our previous statement [1] that "*it has been shown that both phases II and "III" belong to the same superspace group $P2_12_12_1(00\gamma)$* " (with ref. [4] cited as evidence in support of this statement) was misleading, as it was not supported by the published evidence at that time. Nevertheless, we emphasize that any model is based on certain underlying assumptions, and one assumption in the case of our phenomenological model developed in ref. [1] was that phase II and phase "III" have the same superspace group $P2_12_12_1(00\gamma)$. Clearly this assumption, while not proven by published work at the time of publication of ref. [1], is now vindicated by the new data presented here.

However, all other statements in our RSOS paper (ref. [1] of our revised Reply) that refer to our previous EPL paper (ref. [4] of our revised Reply) are unaffected by the uncertainty over whether the data at 147 K represented phase II or phase "III", as these other statements refer to the conclusion that all diffraction maxima for phase II and phase "III" can be indexed on the basis of a single misfit parameter and hence indexed on the basis of a minimal (3+1)-dimensional superspace. This

conclusion from our EPL paper still holds. Indeed, in their recent Comment [2], Toudic et al. do not contest the fact that the diffraction data for both phase II and phase "III" can be indexed on the basis of a minimal (3+1)-dimensional superspace.

Any additional statements they want to include in this comment about subsequent work have to be supported with experimental data that others can judge (e.g. as SI or via link to a preprint of any future publication such as [6]).

As stated above, the revised version of our Reply has been substantially extended to include the main results from our new experiments. The observation of the crucial reflections of the type $(h, k, l, 0)$ with $h+k$ odd in phase II fully support the assignment of the four-dimensional superspace group $P2_12_12_1(00\gamma)$ to phase II. As our phenomenological model (i.e. our paper published in RSOS; ref. [1] of our revised Reply) for rationalization of the phase transitions in this incommensurate material is based on phase II and phase "III" having the same four-dimensional superspace group $P2_12_12_1(00\gamma)$, it follows that the results from our new X-ray diffraction study are in full support of our phenomenological model, and we hope that this point has been fully clarified in our revised Reply.

Appendix B

Dr Laura Smith
Publishing Editor, Journals
Royal Society of Chemistry

Dear Dr Smith

Many thanks for your e-mail received on 24 June informing us that our article (RSOS-190518.R1) will be accepted for publication in Royal Society Open Science subject to minor revision in accordance with the referee suggestions.

Reviewer 1 and Reviewer 2 do not request any further revisions, while Reviewer 3 recommends two changes. Accordingly, we have revised the manuscript to follow the requests of Reviewer 3 as detailed below. Again, we are grateful to Reviewer 3 for their constructive and helpful suggestions.

All changes that have been made are highlighted in green in the revised manuscript.

Comment of Reviewer 3:

P3 L4-6: I don't think that the sentence "Toudic et al do not contest our assertion that the diffraction data for both phase II and phase "III" can be indexed ... (3+1) dimensional superspace" should be included in this Reply. It addresses an issue that wasn't raised in the RSOS comment. The issue was, however, discussed extensively in the earlier EPL comment by Toudic [7] to which Couzi didn't respond. The debate here also concerns the n-hexadecane compound, and no new data is presented on this. P6 L37 should be updated at the same time. Any further discussion on this point should be via regular peer-reviewed publication.

Our Response:

We have deleted the following sentence, as recommended by the reviewer:

"We note that, in their Comment [2], Toudic *et al.* do not contest our assertion that the diffraction data for both phase II and phase "III" can be indexed on the basis of a single incommensurate misfit parameter, and hence can be indexed on the basis of a minimal (3+1)-dimensional superspace."

With regard to the final sentence of the reviewer, we agree that further discussion of the point raised should be made through future peer-reviewed publication, and we will do so in due course.

Comment of Reviewer 3:

P3 L37: I would suggest "vindicated" is changed to "supported". Vindicated, at least in its etymology, can be interpreted as a somewhat aggressive term. As such it doesn't mirror the acknowledgement in the preceding section that statements were made without scientific evidence.

Our Response:

We have changed the word "vindicated" to "supported", as suggested by the reviewer.

Additional Minor Revisions

In addition to the minor revisions in response to Reviewer 3 described above, we have made two additional minor revisions as follows:

(1) Although the authors for correspondence were indicated in the footnote on page 1 of the manuscript, the asterisks used to indicate the correspondence authors were not actually included in the author list. These asterisks have now been added.

(2) We felt it would be appropriate to add the following sentence on page 2, in order to provide a link between the earlier part of the paper in which we discuss the background and content of the Comment of Toudic et al., and the subsequent part of the paper in which we address the issues raised in their Comment:

"In this Reply, we respond to the issues raised in the Comment of Toudic *et al.* [2]."

Please do not hesitate to contact me if you require any additional information.

Yours sincerely,

François Guillaume